# Memory-Efficient Adaptive Optimization

**Rohan Anil**    **Vineet Gupta**
Google Brain
{rohananil,vineet}@google.com

**Tomer Koren**
Google Brain and Tel Aviv Univ.
tkoren@google.com

**Yoram Singer**
Princeton Univ.
y.s@cs.princeton.edu

## Abstract

Adaptive gradient-based optimizers such as Adagrad and Adam are crucial for achieving state-of-the-art performance in machine translation and language modeling. However, these methods maintain second-order statistics for each parameter, thus introducing significant memory overheads that restrict the size of the model being used as well as the number of examples in a mini-batch. We describe an effective and flexible adaptive optimization method with greatly reduced memory overhead. Our method retains the benefits of per-parameter adaptivity while allowing significantly larger models and batch sizes. We give convergence guarantees for our method, and demonstrate its effectiveness in training very large translation and language models with up to 2-fold speedups compared to the state-of-the-art.

## 1   Introduction

Adaptive gradient-based optimizers such as Adagrad [11] and Adam [15] are among the de facto methods of choice in modern machine learning. These methods adaptively tune the learning rate for each parameter during the optimization process using cumulative second-order statistics. Often offering superior convergence properties, these methods are very attractive in large scale applications due to their moderate time and space requirements, which are linear in the number of parameters. However, when training extremely large models even the modest memory overhead imposes grave limitations on the quality of the trained model. For example, recent advances in natural language processing [26, 17] show that models with hundreds of millions to billions of parameters, trained with adaptive optimization methods, achieve state-of-the-art results. In such cases, the memory overhead of the optimizer severely restricts the size of the model that can be used as well as the number of examples in each mini-batch, both of which have a dramatic effect on the accuracy of the model.

Motivated by these challenges, we describe an adaptive optimization method that retains the benefits of standard per-parameter adaptivity while significantly reducing memory overhead. Our construction is general and flexible, and very simple to implement. We give convergence guarantees for our method in the convex (online or stochastic) optimization setting, and demonstrate experimentally that it is particularly effective when the gradients exhibit natural *activation patterns*; namely, when the parameters can be subdivided into (not necessarily disjoint) sets where gradient entries within sets are correlated and of a similar order of magnitude. For example, we often observe in deep networks that the incoming (outgoing) edges into (from) a neuron are jointly activated and, loosely speaking, their associated gradients exhibit similar statistical characteristics. That said, our analysis of the algorithm makes no statistical assumptions on the gradients and is applicable for general stochastic convex optimization. Further, we do not assume that the activation pattern is fully prescribed a-priori.

Large scale experiments show that our algorithm achieves comparable, and at times superior, rates of convergence compared to standard linear-space adaptive methods. Focusing primarily on language modeling tasks where state-of-the-art models are extremely large, we further demonstrate that the reduction in memory footprint can be utilized for a substantial increase in the batch size, which greatly speeds up convergence in a distributed environment. For a fixed budget of computational resource our method is able to shorten the end-to-end walltime for convergence by up to 50%. Our

method exhibits slightly improved per-step time. The latter could be attributed to reduction in the frequency of memory accesses.

## 1.1 Related work

Adaptive learning rates in online and stochastic optimization date back at least to [5] and were popularized in [11, 16], the former of which introduced the well-known Adagrad algorithm. Several variants of Adagrad have now been proposed in the optimization and machine learning literature (see [19] and the references therein), the most notable of which is Adam [15]. All of these methods require (at least) linear space for maintaining various per-parameter statistics during their execution. One notable exception, which is directly related to our work, is the Adafactor algorithm [23] that was proposed as a way to reduce the memory costs of Adam, primarily for training large language models. While the memory requirements of our construction are similar to Adafactor's, the application scope and the convergence properties of the two algorithms are quite different. We discuss the relationship in more detail in Section 4 and give an empirical comparison between the algorithms in Section 5.

Spring et al. [25] provide an alternative way to reduce memory costs, making use of the Count-Sketch data structure [7] to maintain a compressed approximation to the auxiliary variables. One key difference between SM3 and Count-Sketch is that SM3 uses specific hash functions instead of random hash functions. Our hash functions are compatible with slices of parameter tensors and are geared towards exploiting empirically observed correlations between the auxiliary parameters, as we discuss below (see Section 4). As a result, our sketches can be 100x–1000x smaller than the original tensors—compared to the 5x reduction reported in [25]—while showing significantly smaller approximation error (we provide details in the full version of the paper [3]). In addition, randomized sketching is extremely inefficient to implement on GPUs and TPUs, since it involves sparse look-ups and is not cache-efficient. These differences allow us to show significant improvements for a large variety of tasks and models, as compared to the results in [25].

Also related to our work is the Shampoo algorithm for optimization over tensor structures [12]. The goal of Shampoo is very different from ours: going beyond entry-wise learning rates and employing *full-matrix* regularization in a computationally efficient way. Nonetheless, Shampoo can also be seen as a method to substantially reduce the memory footprint of full-matrix preconditioned algorithms (specifically, full-matrix Adagrad). In a sense, our algorithms are analogous to a diagonalized version of the Shampoo algorithm. Yet another recent adaptive optimization method is the GGT algorithm [2]. Similarly to Shampoo, the goal of the latter is to reduce the computation cost of full-matrix preconditioning in order to make it practical in large scale settings. However, GGT stores multiple copies of the gradient over the course of its execution, and as a result, its space requirements restricts it from being applied at large scale.

## 2 Preliminaries

### 2.1 Online optimization

We henceforth assume the general *online optimization* setting (see [22, 13]). Online optimization consists of rounds $t = 1, \ldots, T$, where in each round the algorithm chooses a parameter vector $w_t \in \mathbb{R}^d$. After making a choice on round $t$, the algorithm receives a loss function $\ell_t : \mathbb{R}^d \to \mathbb{R}$ which is used to form an update of the parameters. In our analysis, we focus on online *convex* optimization in which $\ell_1, \ldots, \ell_T$ are convex. Often, as is the case in this paper, the update is determined by the gradient $g_t = \nabla \ell_t(w_t)$ of the instantaneous loss $\ell_t$ at the current iterate $w_t$. The algorithm is measured by its $T$-round regret with respect to a given comparator $w^\star \in \mathbb{R}^d$, defined as the quantity $\sum_{t=1}^{T} \ell_t(w_t) - \sum_{t=1}^{T} \ell_t(w^\star)$. An online optimization algorithm is convergent if its regret is $o(T)$, i.e., its average regret approaches zero as $T$ grows.

The above setting includes stochastic (possibly mini-batched) optimization as a special case. In stochastic optimization the underlying goal is to minimize a population loss $L(w) = \mathbb{E}_{z \sim D}[\ell(w, z)]$ based on samples of $z$. Here $\ell(w, z)$ defines the loss of parameters $w$ w.r.t a batch $z$. The online loss function $\ell_t(w) = \ell(w, z_t)$ is the average loss over a mini-batch $z_t$ received on iteration $t$. The stochastic gradient $g_t$ is a conditionally unbiased estimate of the gradient of $L$ at the current parameter vector $w_t$. Under convexity assumptions, an online algorithm with vanishing average regret can be converted to a stochastic optimization algorithm for minimizing the population loss $L$ [6].

## 2.2 Adaptive methods

For the sake of self-containment, we give a brief description of adaptive gradient methods, focusing on Adagrad [11]. Adagrad maintains at each step $t$ parameter-wise accumulated statistics which are computed from the previously obtained gradients $g_1, \ldots, g_t$:

$$\gamma_t(i) = \sum_{s=1}^{t} g_s^2(i), \qquad \forall i \in [d]. \tag{1}$$

Based on these statistics, the update rule of the algorithm on step $t$ takes the form:

$$w_{t+1}(i) = w_t(i) - \eta \frac{g_t(i)}{\sqrt{\gamma_t(i)}}, \qquad \forall i \in [d],$$

where $\eta > 0$ is an external learning rate parameter. Duchi et al. [11] proved the following regret bound for Adagrad with respect to a given $w^\star$ (with properly tuned $\eta$):

$$\sum_{t=1}^{T} \ell_t(w_t) - \sum_{t=1}^{T} \ell_t(w^\star) = O\left( D \sum_{i=1}^{d} \sqrt{\sum_{t=1}^{T} g_t^2(j)} \right), \tag{2}$$

where $D \geq \max_t \|w_t - w^\star\|_\infty$. Adagrad has proved to be particularly useful in training sparse models, where the effective learning rates $\eta/\sqrt{\gamma_t(i)}$ decay in a moderate way for rare, yet potentially informative, features. In these settings, Adagrad can potentially lead to substantial improvements in convergence time; see for instance the discussion in [11]. Crucially, however, Adagrad must maintain auxiliary sequence of accumulators $\gamma_t$ and thus needs $\Omega(d)$ additional space. The goal of this paper is to provide memory-efficient methods with comparable convergence characteristics that refrain from maintaining the full vectors $\gamma_t$.

## 3 The SM3 Algorithm

We now present our memory-efficient adaptive optimization algorithm. As an abstraction, the algorithm employs a *cover* of the parameters: a collection of $k$ nonempty sets $\{S_r\}_{r=1}^{k}$, such that $S_r \subseteq [d]$ and $\cup_r S_r = [d]$. In particular, each index $i \in [d]$ may be contained in multiple sets $S_r$. The algorithm maintains a single variable for each set $S_r$ in the cover. Thus, the additional space it requires is $O(k)$ rather than the $O(d)$ required by standard adaptive methods. In large scale applications, $k$ will be chosen to be negligible in comparison to $d$, which would translates to substantial savings in memory; see Section 4 for a discussion on the covers used in practice.

Concretely, for each set $S_r$ in the cover, the algorithm maintains a running sum, $\mu_t(r)$, of the *maximal* variance over all gradient entries $j \in S_r$. Next, for each parameter $i$, we take the *minimum* over all variables $\mu_t(r)$ associated with sets which cover $i$, denoted $S_r \ni i$. Thereafter, the learning rate corresponding to the $i$'th gradient entry is determined by taking the square-root of this minimum, denoted by $\nu_t(i)$. Accordingly, we name our algorithm the *S*quare-root of *M*inima of *S*ums of *M*axima of *S*quared-gradients *M*ethod, or in short, *SM3*. See Algorithm SM3-I for its pseudocode.

As noted above, SM3-I requires only $O(k)$ space in addition to the space required for storing the parameters $w_t$ themselves. The time per iteration of SM3-I is $O(\sum_{r=1}^{k} |S_r|)$. To see this, consider a bipartite graph defined over $d+k$ vertices. Nodes on one side of the graph correspond to indices $i \in [d]$, while nodes on the other side correspond to indices $j \in [k]$. The edges of the graphs are all pairs $(i,j)$ such that $i \in S_j$. The complexity of each inner for-loop of the algorithm scales with the number of edges in this graph, which is equal to $O(\sum_{r=1}^{k} |S_r|)$. Note that updating the weights $w_t$ takes $O(d)$ time, which is always dominated by the former quantity.

---

**SM3-I**

1: **parameters:** learning rate $\eta$
2: initialize $w_1 = 0$ ; $\forall r \in [k] : \mu_0(r) = 0$
3: **for** $t = 1, \ldots, T$ **do**
4:     receive gradient $g_t = \nabla \ell_t(w_t)$
5:     **for** $r = 1, \ldots, k$ **do**
6:         set $\mu_t(r) \leftarrow \mu_{t-1}(r) + \max_{j \in S_r} g_t^2(j)$
7:     **for** $i = 1, \ldots, d$ **do**
8:         set $\nu_t(i) \leftarrow \min_{r:S_r \ni i} \mu_t(r)$
9:         update $w_{t+1}(i) \leftarrow w_t(i) - \eta\, g_t(i)/\sqrt{\nu_t(i)}$
                  ▷ with the convention that $0/0 = 0$

---

The following provides convergence guarantees for SM3-I.

**Proposition 1.** *Assume that the loss functions $\ell_1, \ell_2, \ldots$ are convex, and let $w_1, w_2, \ldots$ be the iterates generated by SM3-I. Then, for any $w^\star \in \mathbb{R}^d$,*

$$\sum_{t=1}^{T} \big(\ell_t(w_t) - \ell_t(w^\star)\big) \le 2D \sum_{i=1}^{d} \sqrt{\min_{r:S_r \ni i} \sum_{t=1}^{T} \max_{j \in S_r} g_t^2(j)} \,,$$

*where $\max_t \|w_t - w^\star\|_\infty \le D$ and choosing $\eta = D.$*[1]

For stochastic optimization, i.e., when the functions $\ell_t$ correspond to i.i.d. samples with $\mathbb{E}[\ell_t(w)] = L(w)$, the above bound translates via standard arguments to a $O(1/\sqrt{T})$-type convergence guarantee for the average iterate $\overline{w}_T = \frac{1}{T} \sum_{t=1}^{T} w_t$ of the form

$$\mathbb{E}[L(\overline{w}_T)] - L(w^\star) = O\left( \frac{1}{T} \sum_{i=1}^{d} \mathbb{E} \sqrt{\min_{r:S_r \ni i} \sum_{t=1}^{T} \max_{j \in S_r} g_t^2(j)} \right).$$

Note that adding more sets $S_r$ to the cover used by SM3 always improves its convergence bound, but results in a worse space complexity and a higher runtime per step. When $k = d$ and $S_i = \{i\}$ for all $i \in [d]$, SM3-I reduces to the Adagrad algorithm, and the regret bound in Proposition 1 then precisely recovers the bound attained by Adagrad (recall Eq. (2)). In general, the right-hand side of Proposition 1 is never smaller than Adagrad's regret bound, as expected from a space-restricted scheme (this is a consequence of Claim 2 below). Nevertheless, the two bounds can be of similar order of magnitude in practical scenarios; see Section 4 below for a detailed discussion.

We now give a proof of Proposition 1. First, we state two elementary properties of the step sizes the algorithm computes. For a proof, see the full version of the paper [3].

**Claim 2.** *For any $i$, the sequence $\nu_1(i), \nu_2(i), \ldots$ is monotonically increasing, and $\nu_t(i) \ge \sum_{s=1}^{t} g_s^2(i)$.*

*Proof of Proposition 1.* Let us first assume that $g_1(i) > 0$ for all $i$, so that $\nu_t(i) > 0$ for all $i$ and $t \ge 1$ due to Claim 2. We start by observing that SM3-I performs Online Mirror Descent updates, where the step on round $t$ uses the positive definite diagonal matrix $H_t = \mathrm{diag}(\nu_t^{1/2})$ for regularization. Then, employing a standard regret bound for the Online Mirror Descent algorithm with time-dependent regularization (see for instance [11, Proposition 3]), the regret of the algorithm is bounded by

$$\frac{1}{2\eta} \sum_{t=1}^{T} \big( \|w_t - w^\star\|_{H_t}^2 - \|w_{t+1} - w^\star\|_{H_t}^2 \big) + \frac{\eta}{2} \sum_{t=1}^{T} \big( \|g_t\|_{H_t}^* \big)^2 \,.$$

Here, $\|x\|_H = \sqrt{x^\mathsf{T} H x}$ and $\|\cdot\|^*$ is the corresponding dual norm, $\|x\|_H^* = \sqrt{x^\mathsf{T} H^{-1} x}$. Henceforth, for notational convenience we set $\nu_0 = 0$. Simplifying the first sum above using the fact that $H_t$ are diagonal matrices, we have

$$\sum_{t=1}^{T} \big( \|w_t - w^\star\|_{H_t}^2 - \|w_{t+1} - w^\star\|_{H_t}^2 \big) \le \sum_{t=1}^{T} (\nu_t^{1/2} - \nu_{t-1}^{1/2}) \cdot (w_t - w^\star)^2$$

$$\le \sum_{t=1}^{T} (\nu_t^{1/2} - \nu_{t-1}^{1/2}) \cdot \big( \|w_t - w^\star\|_\infty^2 \mathbf{1}_d \big)$$

$$\le D^2 \big( \nu_T^{1/2} \cdot \mathbf{1}_d \big) \;=\; D^2 \, \mathrm{Tr}(H_T) \,.$$

Now, let $\gamma_t(i) = \sum_{s=1}^{t} g_s^2(i)$ and consider the positive definite diagonal matrix $G_t = \mathrm{diag}(\gamma_t^{1/2})$. From [12, Lemma 2] with $\Phi(G) = \mathrm{Tr}(G)$, we have

$$\sum_{t=1}^{T} \big( \|g_t\|_{G_t}^* \big)^2 \le \sum_{t=1}^{T} \big( \|g_t\|_{G_T}^* \big)^2 + \mathrm{Tr}(G_T) = \gamma_T^{-1/2} \cdot \gamma_T + \mathrm{Tr}(G_T) = 2 \, \mathrm{Tr}(G_T) \,.$$

Also, from Claim 2 we know that for all $t$, $H_t \succeq G_t$, thus

$$\sum_{t=1}^{T} \left( \|g_t\|_{H_t}^* \right)^2 \le \sum_{t=1}^{T} \left( \|g_t\|_{G_t}^* \right)^2 \le 2 \operatorname{Tr}(G_T) \le 2 \operatorname{Tr}(H_T) \,.$$

In summary, we have established that

$$\sum_{t=1}^{T} \ell_t(w_t) - \ell_t(w^\star) \le \left( \frac{D^2}{2\eta} + \eta \right) \operatorname{Tr}(H_T) \,.$$

Plugging in $\eta = D$ and the expression for the diagonal elements of $H_T$, we obtain the claim.

For the degenerate case where the matrices $H_t$ may not be strictly positive definite, a careful yet technical inspection of the proof above reveals that our arguments apply to this case as well by replacing inverses with pseudo-inverses. The rest of the proof remains intact as the algorithm does not update parameter $i$ on step $t$ if the corresponding diagonal entry in $H_t$ is zero. $\square$

## 3.1 SM3-II

We now discuss a slightly more efficient variant of SM3, which we describe in SM3-II. It is similar to SM3-I, and improves on the latter in the following sense.

**Proposition 3.** *For any $i \in [d]$, the sequence $\nu_1'(i), \ldots, \nu_T'(i)$ is monotonically increasing. Further, fixing a sequence of gradients $g_1, \ldots, g_T$, we have for all $t, i$ that $\sum_{s=1}^{t} g_s^2(i) \le \nu_t'(i) \le \nu_t(i)$, where $\nu_1(i), \ldots, \nu_T(i)$ is the sequence SM3-I emits upon receiving the gradients $g_1, \ldots, g_T$.*

(See the full version of the paper [3] for a proof.) In other words, SM3-II provides a tighter upper bound on the cumulative gradient squares than SM3-I. Consequently, we can show, along similar lines to the proof of Proposition 1, a slightly better bound for SM3-II that scales with the quantity $\sum_{i=1}^{d} \sqrt{\nu_t'(i)}$, which is always smaller than the one appearing in the bound of SM3-I.

---

**SM3-II**

1: **parameters:** learning rate $\eta$
2: initialize $w_1 = 0$ ; $\forall r \in [k] : \mu_0'(r) = 0$
3: **for** $t = 1, \ldots, T$ **do**
4:     receive gradient $g_t = \nabla \ell_t(w_t)$
5:     initialize $\mu_t'(r) = 0$ for all $r \in [k]$
6:     **for** $i = 1, \ldots, d$ **do**
7:         $\nu_t'(i) \leftarrow \min_{r:S_r \ni i} \mu_{t-1}'(r) + g_t^2(i)$
8:         $w_{t+1}(i) \leftarrow w_t(i) - \eta \, g_t(i) / \sqrt{\nu_t'(i)}$
            $\triangleright$ with the convention that $0/0 = 0$
9:         **for** all $r : S_r \ni i$ **do**
10:             $\mu_t'(r) \leftarrow \max\{\mu_t'(r), \nu_t'(i)\}$

---

## 4 Discussion

Thus far, we gave an analysis of SM3 in a worst-case (convex) setting without placing any further assumptions on the statistical characteristics of the underlying stochastic gradients. Further, we did not attempt to relate the cover used by SM3 to properties of the underlying stochastic optimization problem. It should not come as a surprise that in this general setting, the convergence of SM3 might be much worse, at least in theory, than its linear-memory counterpart Adagrad.

**Activation patterns.** Often in our experiments, we observe common statistical attributes that could be exploited by SM3. Specifically, we see that certain entries of the stochastic gradients have (on average) similar values, and exhibit what we refer to as an *activation pattern*. For example, in gradients of embedding layers of deep networks, an entire row (or column) is either zero or non-zero. Similarly, in intermediate layers we often observe that gradients associated with the same unit are of similar order of magnitude. In these cases, a similar phenomenon is observed in the second-order statistics maintained by adaptive methods. In Figure 1 we visualize this phenomenon for different layers of a Transformer network. In the full version of the paper [3] we give additional illustrations of similar phenomena in convolutional layers of image classification models.

**Choice of covers.** The intuitive notion of an activation pattern motivates a natural and generic choice for the cover used by SM3 in practice. For the parameters of deep networks, that are organized as a collection of tensors, we form a cover consisting of slices of co-dimension 1 for each tensor. Thus, for an $m \times n$ parameter matrix, the cover consists of rows and columns of the matrix. The memory requirements therefore drop from $\Theta(mn)$ to merely $\Theta(m + n)$. For a parameter tensor of dimension $n_1 \times \cdots \times n_p$, the reduction in memory consumption is even more pronounced, dropping

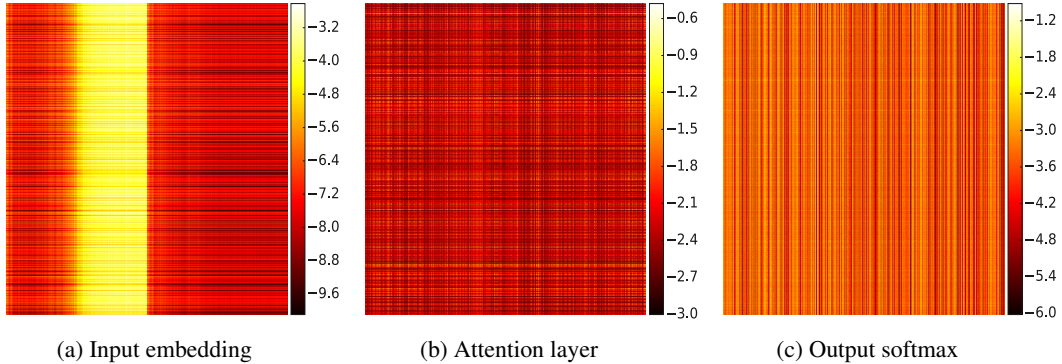

| (a) Input embedding | (b) Attention layer | (c) Output softmax |

Figure 1: Visualization of Adagrad's statistics (cf. Eq. (1)) for different weight matrices in Transformer-Big model trained with Adagrad on WMT'14 en→fr (color intensities are in log scale).

from $\Theta(\prod_{i=1}^{p} n_i)$ to $\Theta(\sum_{i=1}^{p} n_i)$. This virtually eliminates the memory overhead associated with maintaining the adaptive learning rates.

We argue, though only informally, that when choice of cover used by SM3 is compatible with the observed activation patterns, we expect the convergence of SM3 to be significantly better, and closely match Adagrad. Quantitatively, if each parameter $i \in [d]$ is covered by a set $S_r$ such that $g_s(j) \approx g_s(i)$ for all $j \in S_r$, then $\max_{j \in S_r} g_s^2(j) \approx g_s^2(i)$, and thus $\min_{r:S_r \ni i} \sum_s \max_{j \in S_r} g_s^2(j) \approx \sum_s g_s^2(i)$. Thus, the bounds in Proposition 1 and Eq. (2) are of similar order of magnitude. In other words, in such scenarios we inherit the convergence properties of Adagrad while using a negligible amount of memory. We remark that the activation pattern need not be fully specified in advance; in particular, SM3 is robust to whether a certain parameter is "row tied" or "column tied", as long as both rows and columns are included in the cover.

**Comparison with Adafactor.** Adafactor [23] is a very effective method for space-efficient adaptive optimization. SM3 and Adafactor differ in a number of important ways. First, Adafactor is only defined for matrix-shaped parameters while SM3 applies to tensors of arbitrary dimensions, and even more generally, to any predefined cover of the parameters. Second, Adafactor is in essence a fixed learning-rate algorithm, being a memory-constrained variation of Adam, and often requires a manually devised learning-rate schedule to ensure convergence. In contrast, SM3 adapts its learning rates in an adaptive, data-driven manner similar to Adagrad. Finally, SM3 comes with rigorous convergence guarantees in stochastic convex optimization settings.

## 5 Experiments

We demonstrate the practical efficacy of SM3 on several machine learning tasks using published state-of-the-art architectures. We focus on three domains: machine translation, language modeling, and image classification. We implemented SM3 as an optimizer in TensorFlow [1]; source code is publicly available at [4]. Our implementation follows the pseudocode of SM3-II, as it performed slightly yet consistently better than SM3-I in our experiments (as predicted by our bounds). We use covers induced by rows and columns of matrices, and more generally, by slices of higher-order tensors (e.g., in convolutional layers represented by 4-dimensional tensors), as described in Section 4. In addition to being compatible with the natural activation patterns, these covers facilitates efficient tensor operations available on GPUs and TPUs for computing *max* and *min* over the sets. In all experiments, we used the Cloud TPU-v2 device [14] where each core has 8GiB of memory. For more details on all of our experiments, including the precise hyperparameters used in each of them, refer to the full version of the paper [3].

### 5.1 Machine translation

We experimented with machine translation tasks on two standard datasets from WMT'14: English to French (en→fr) with 36.3M sentence pairs, and English to German (en→de) with 4.5M sentence pairs. We used the state-of-the-art Transformer architecture Vaswani et al. [26]. The basic version of this

model has 93.3M parameters and consumes 0.36GiB memory. The larger variant (Transformer-Big) has 375.4M parameters (1.432GiB) and consists of 6 layers for its encoder and decoder, where each layer has 1024 model dimensions, 8192 hidden dimensions, and 16 attention heads.

Here we report our results on the larger Transformer-Big, and defer results on the basic Transformer to the full version of the paper [3]. We trained Transformer-Big on the en→fr dataset with batches of size 384, and compared SM3 with several standard optimizers in each of the tasks. In all cases, we used momentum (including for Adagrad) and extensively tuned all hyperparameters. We also ran SGD with momentum (with various exponential decay schedules), but it performed poorly and hence it is omitted from the figures. The results are provided in Figure 2 and Table 1, and demonstrate that SM3 performed substantially better and provided a large improvement in BLEU score compared to Adam and Adafactor. In addition, the small memory requirements of SM3 and Adafactor allowed us to *double the number of examples in a batch* to a total of 768, with minimal additional computation resources. In this setting, we found that SM3 outperformed Adafactor in terms of the number of steps as well as the wall-time to convergence by roughly a factor of 2. We further observed that SM3 approximated the 2nd-order statistics tightly. For more details, see the full version of the paper [3].

Both models were trained on a $4 \times 4$ Cloud TPU-v2 using the Lingvo [24] sequence modeling framework, with 32K word-pieces [21] for each language pair. BLEU scores were computed on the Newstest 2014 for evaluation, on tokenized, true-case outputs, and without manual post-processing of the text, similar to [28]. Our BLEU scores are not directly comparable to those of [26]. We instead followed the experimental protocol described in a later work [8].

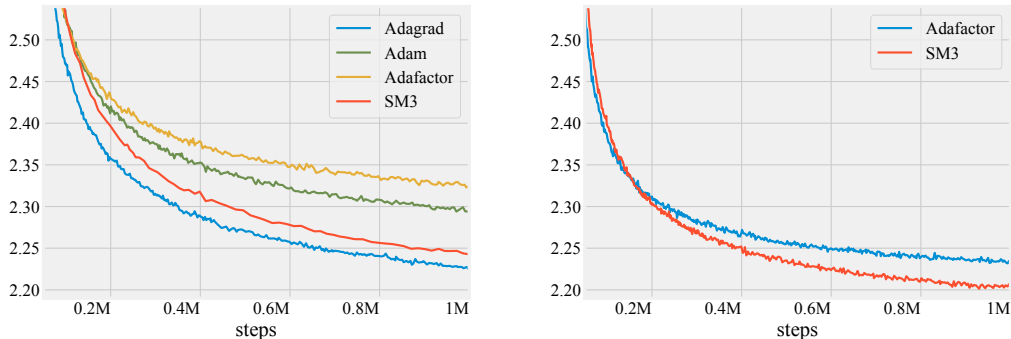

Figure 2: Test log-perplexity of a Transformer-Big model on WMT'14 en→fr, when training with batch sizes of 384 (left) and 768 (right). For batch size of 768, Adam and Adagrad were infeasible as they exceeded the available memory.

| OPTIMIZER | BATCH SIZE PER CORE (TOTAL) | MEMORY USAGE PER CORE | BLEU |
|---|---|---|---|
| Adam | 12 (384) | 6.88 GiB | $38.96 \pm 0.002$ |
| Adagrad | 12 (384) | 6.85 GiB | $39.90 \pm 0.003$ |
| Adafactor | 12 (384) | 5.43 GiB | $37.89 \pm 0.002$ |
| SM3 | 12 (384) | 5.36 GiB | $39.81 \pm 0.002$ |
| Adafactor | 24 (768) | 7.04 GiB | $39.65 \pm 0.002$ |
| SM3 | 24 (768) | 7.02 GiB | $\mathbf{40.50 \pm 0.001}$ |

Table 1: BLEU scores and memory usage for various batch sizes on the WMT'14 en→fr dataset.

## 5.2 Language modeling

Next, we considered a language modeling task on the concatenation of Wikipedia and BooksCorpus [29], with 2.5B and 800M words respectively. We used the recent Bidirectional Encoder Representation (BERT) architecture of Devlin et al. [10], focusing on its larger variant, coined BERT-Large. BERT-Large is a large bidirectional transformer model containing 24 transformer blocks with 1024 hidden dimensions and 16 self attention heads. It has 340M parameters (1.297 GiB), and is set up to jointly optimize two objectives: (a) masked language model (Masked-LM) loss where the task is to

| OPTIMIZER | BATCH SIZE PER CORE (TOTAL) | MEMORY USAGE PER CORE |
|---|---|---|
| Adam | 8 (1024) | 6.15 GiB |
| SM3 | 8 (1024) | 4.90 GiB |
| SM3 | 16 (2048) | 6.02 GiB |

Table 2: Training memory consumption at different batch sizes for BERT-Large on 8x8 TPUs.

predict masked tokens based on surrounding context, and (b) next sentence prediction (NSP) loss where the task is to predict whether two given sentences are consecutive in the text.

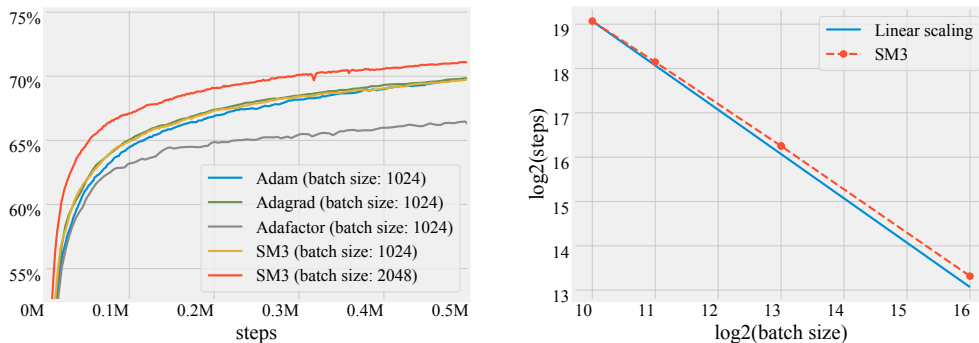

Figure 3: Masked LM test accuracy (left), and number of steps to get 70% test accuracy as a function of the batch size (right), of the BERT-Large language model trained on Wikipedia+BooksCorpus. SM3 with batch size 2048 uses about the same amount of memory as Adam/Adagrad with batch size 1024, and scales linearly up to a batch size of $2^{16}$, at which point we hit the hardware memory limits.

As before, we compared SM3 with Adagrad, Adam and Adafactor. Our results are presented in Figure 3. We see that SM3 worked as well as Adam and Adagrad for a fixed batch size. However, the savings in memory allowed us to train SM3 with double the batch size, resulting in a substantial increase in accuracy. The experiments were run using the open sourced code from [10] on a $8 \times 8$ Cloud TPU-V2 configuration.

To underscore the importance of our memory savings in the context of very large models, we report additional results on the number of steps required for reaching a given solution quality for various batch sizes. We chose a solution quality of 70% Masked-LM accuracy on the holdout set, which Adam and AdaGrad reached at 500k steps. We use Cloud TPU-v3 device which has 16Gib per core for this experiment. We measured the number of steps SM3 needed to reach this accuracy as a function of the batch size. Our results are presented in Figure 3. SM3 scaled almost linearly with the batch size, up to a size of $2^{16}$, at which point the training program reached the limits of memory available on hardware. We also found that SM3 came out ahead in terms of wall-time: with the same batch size, a step of SM3 was faster than Adam's by 3%, and doubling the batch size allowed it to reach the same solution quality in almost 35% less wall-time for the same computational budget.

### 5.3 AmoebaNet-D on ImageNet

Finally, we report results from a different domain: image classification on ImageNet [20] with the state-of-the-art AmoebaNet-D architecture [18], that has recently won the Stanford DAWNBench competition [9]. We compared SM3 with SGD with momentum (Adam performed poorly on this task). SM3 performed very well in this task and achieved improved convergence to state-of-the-art performance, reaching 78.71% top-1 and 94.31% top-5 test accuracies. The fully detailed convergence plots are provided in the full version of the paper [3].

## 6 Summary

Motivated by the large increase in models sizes and the huge amounts of memory required for training them, we have presented a new memory-efficient adaptive optimization algorithm for stochastic optimization called SM3. We demonstrated empirically that SM3 can be used effectively in training

modern mammoth-sized models and dramatically decrease memory overhead. Utilizing the freed memory for increasing the batch size, our experiments indicate that this saving can also lead to significant improvements in performance. Our theoretical investigation focused on convex objectives. As with many other optimization scenarios, we believe the analysis of convex memory-efficient adaptive optimization could serve as a basis for understanding non-convex settings.

Our memory savings virtually eliminate the overhead coming from the second-order statistics $\gamma_t$ with little and often no impact on convergence. Additional and potentially substantial improvements in memory consumption could come from compressing or sketching the momentum terms employed by virtually all first-order optimizers used in practice. We leave the exploration of this promising direction for future work.

## Acknowledgements

We would like to thank Luke Metz, Kunal Talwar and Yonghui Wu for numerous helpful discussions and suggestions. Special thanks go to Samy Bengio who made it possible for us to conduct large scale experiments on a tight schedule. We would also like to thank Zhifeng Chen for coming up with the shorthand 'SM3'.

## Footnotes

[1]Here we implicitly assume that the iterates of the algorithm remain bounded and $D$ is a constant. This can be enforced by projecting the iterates to a bounded set of choice; we avoid introducing projections explicitly as they are rarely used in practice.

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
