[Reviews · NeurIPS 2019]

Reviewer 1



The paper addresses an important problem in deep learning. As larger-and-larger models are being trained on increasing amounts of data, on-device memory becomes a bottleneck. Efforts which reduce storage of auxiliary information including gradient moments and activations, allow for larger models and/or batches on same hardware. I endorse the paper for acceptance to NeurIPS. The paper is well-written and the results are impressive. I especially appreciate that the authors tested the algorithm on cutting-edge models such as ImageNet, BERT and Transformer MT, and also had an ablation on hyperparameters such as batch sizes (Figure 3b). There are some ways that the paper can be improved: 1. The algorithm section can be improved a bit. As it stands, the algorithm is written a bit too abstractly/generally. This is impeding readability, especially about the covering sets. I'd highly suggest that the authors reduce the abstraction and (for instance) present the algorithm for simple deep networks and then qualifying how this can be generalized. 2. I would have liked a deeper comparison between Adafactor and SM3. The authors do discuss it at the end of Section 4, however, it is unclear from simply reading that paragraph what the differences would be between the two for simple use-cases such as Transformers, ConvNets or Feed-Forward networks. 3. I was hoping that the code for SM3 was included with the paper. It was not. I would encourage the authors to include or open-source that if possible. 4. It is not totally clear what the strategy is with learning rates. The authors state that their algorithm is unlike Adafactor in that it decays similar to Adagrad rather than being somewhat constant (similar to Adam). Do you reduce the learning rate as you go along? Do you scale the learning rates linearly with batch sizes? Do you warmup the learning rates? POST REBUTTAL EDIT : I have read through the rebuttal and the other reviews and maintain that this paper should be accepted. I thank the authors for clarifying my questions.

Reviewer 2



Review Update: "For this reason, SKMS’19 only apply their techniques to sparse (sampled) softmax and embedding layers with block versions of the hashes. They leave the treatment of hidden layers to future work." This statement is incorrect. The ICML version of the paper showed results compressing all of the layers of the neural network for ResNet-18 and Transformer-XL.The paper showed how to adapt count-sketches for GPUs and did not show significant computational slow-down compared to the full-sized baseline. Their code is publically available for PyTorch. "Space saving: SKMS’19 uses sketches that are 5x smaller than the original tensors, whereas our sketches are 100x-1000x times smaller." It is important to compare how much overall GPU or TPU memory is saved rather than the data structures themselves. Your paper didn't demonstrate 100x-1000x overall memory savings. i.e. From Table 2, Adam 6.15 GB => SM3 4.9 GB (Overall Savings - ~20% savings). Those saving are on-par with those reported in the SKMS'19 paper. "We could not obtain reasonable results with SKMS’19 (and they do not report such results)." Their paper reported comparable results against SGD on ImageNet with ResNet-18 and their source code is publically available. I don't see why it wouldn't perform as well as SM3 if you used the AmoebaNet-D architecture. I also have new several questions about the compression quality experiment in Figure 1 of the rebuttal. 1. Is the error in the count-sketch significant enough to hurt the convergence rate and performance of the optimizer? 2. At what time step in the optimization process, did you perform this comparison? i.e. iteration 1, 1000, 1,000,000 3. Does this gap hold throughout the optimization process? 4. How much memory was allocated for the count-sketch? What are the dimensions of the count-sketch? 5. Why did you select only the top 100 parameters? A single neural network layer has millions of parameters. The author's algorithm and experiments are geared towards Tensorflow and TPUs, while SKMS'19 worked with PyTorch and GPUs. Algorithmically, SM3 is a modified-version of the count-sketch data structure. The rebuttal did refute how you could not modify the count-sketch data structure to behave like the SM3 algorithm. ------------------------------------------------------------------------- -- Summary -- Modern adaptive optimizers require additional parameters that grow linearly with the number of parameters. This constant memory overhead causes problems with large-scale training. This paper proposes an adaptive optimization method that retains per-parameter adaptivity while reducing memory overhead. The additional memory savings is used for either a larger batch-size or more expressive model. For a tensor [N_1, N_2, ..., N_p] where d = prod(N_1, N_2, ..., N_p). Each parameter i in [d] is associated with p variables. A variable for each dimension of the tensor. All parameters in the same tensor dimension (i.e. row or column) share the same variable. For parameter i, we take the minimum value among the p variables. -- Originality -- In "Compressing Gradient Optimizers via Count-Sketches" (ICML 2019), the paper proposed using the count-min sketch (CMS) data structure to compress the additional parameters to reduce the optimizer's memory overhead, which is the same objective as this paper. There are clear similarities with the cover set and the count-min sketch (CMS): 1. For the count-min sketch, parameters are randomly mapped to p variables. For the cover set, parameters deterministically mapped along the tensor's dimensions to p variables. 2. In both approaches, the minimum value among p variables is used as the parameter's estimate. 3. The count-min sketch aggregates all updates, while SM3-I takes the maximum value from all parameters in the cover set.

Reviewer 3



I have read the rebuttal and other reviewers' comments. I tend to accept this paper. As to the breakdown of memory usage, I was wondering what is the composition of the (memory) working set during the model training. In practice, when some in-place operations are not available or practical for certain hardware architectures, some additional copies of temporary tensors for computation might be needed. So how would this affect the adoption of SM3? As to the over-parameterization, please check the following papers to see a concrete assumption and its effect on simplifying and improving the convergence analysis. Mass: an accelerated stochastic method for over-parametrized learning Fast and Faster Convergence of SGD for Over-Parameterized Models and an Accelerated Perceptron ---------------------------- This work presents a practical solution (SM3) for reducing the memory space used by some adaptive optimization methods. The proposed solution trades some convergence speed (since it uses smaller learning rate according to the maximum) for some memory space dedicated for maintaining cumulative gradient information. This solution is applicable to AdaGrad-alike methods which uses second moments of the gradients to adapt learning rate during training. The authors show the convergence of SM3 and demonstrate its capability of memory saving without performance degradation on training several important neural networks. I like the proposed method, which is simple and practical. The idea behind is also likely applicable in many other settings. However, I tend to give this paper a borderline score for concerns on novelty. Specifically, the techniques used by Adafactor and SM3 are similar. The second claim in the comparison to Adafactor (end of the section 4) is misleading. Just like Adam, the learning rate of Adafactor should take into account the second moment of the gradients, which does change a lot at the beginning of the training. It is also hard to tell why the technique used by Adafactor cannot be easily applied/extended to high-order tensors. In regard to the experiments, I wish that the authors could give a detailed breakdown on the memory usage. In addition, I would appreciated a deeper discussion on the memory saving with SM3, e.g., what kind of architecture fits SM3 better and what do not? As to the theory, it might be worthy adding more assumptions such like over-parametrization (or interpolation) for better convergence bounds.

[Author Response · NeurIPS 2019]

**Reviewer 1:** Thank you for the clear guidance on how to improve our presentation. We will follow it closely for the final version. That said, your review reflects much more enthusiasm than your score suggests. We would greatly appreciate if your could re-examine your numerical evaluation. ● *Relation to AdaFactor:* Please see response to Reviewer 3. ● *Open-sourcing:* Certainly! We have been finalizing an open-source version to be available soon on GitHub. ● *Learning Rates (LR):* All algorithms we experimented with needed LR warm-up for improved performance. Beyond this, SM3 decays its LRs autonomously (like AdaGrad) and does *not* equire an "external" LR schedule. In contrast, Adam and AdaFactor do rely on an external LR schedules. Full details on the LRs and schedules used are provided in the supplementary. We will clarify this in the final version, thank you for pointing it out!

**Reviewer 2:** Thank you for the constructive comments. It appears that you are pleased by the high significance of the contributions, but have some concerns about novelty which we now address. ● *Relation to "Compressing Gradient Optimizers" (SKMS'19):* First, we note that SKMS'19 should more fairly be seen as a concurrent work to ours rather than a prior work. (The first version of our paper appeared online before the first version of SKMS'19.) This note aside, SM3 is superior to the Count Sketch algorithm of SKMS'19 in several important ways:

(1) *Efficiency:* Randomized sketching is extremely inefficient on GPUs and TPUs as it requires sparse look-ups and is not cache-efficient. For this reason, SKMS'19 only apply their techniques to sparse (sampled) softmax and embedding layers with block versions of the hashes. They leave the treatment of hidden layers to future work. In contrast, our technique is deterministic and cache-friendly and "compresses" *all layers* of the model.
(2) *Space saving:* SKMS'19 uses sketches that are 5x smaller than the original tensors, whereas our sketches are **100x-1000x times smaller**.
(3) *Compression quality:* Empirically, the SM3 compression results in significantly smaller error compared to SKMS'19, as illustrated in the figure below. We plan to include these comparisons in the final version.
(4) *Empirical performance:* These differences allow us to show improvements on a large variety of tasks & models. SKMS'19 essentially brings no improvement for Imagenet, and only has improvements for models dominated by the embedding and softmax layers (along with sampled softmax). In contrast, SM3 converges faster for Imagenet, and for language models such as a state-of-the-art 24-layer BERT, we show significant improvements in convergence. We could **not** obtain reasonable results with SKMS'19 (and they do not report such results).

● *Applying to Adam:* SM3 can be used with exponential moving-averages (like Adam) instead of sums (like Adagrad) through a simple modification of $\nu'_t(i)$ (SM3-II): $\nu'_t(i) \leftarrow \beta \min_{r:S_r \ni i} \mu'_{t-1}(r) + (1 - \beta)g_t^2(i)$. As we discuss in the paper, in our experiments moving-averages performed substantially worse than sums (regardless of memory savings).

**Reviewer 3:** ● *LR schedules, compared to AdaFactor:* The discussion in Section 4 was meant to underscore that the LR in Adam/AdaFactor does not necessarily decay with time, and so practitioners often incorporate manually-tuned external decay schedules (see Table 4 in the supplementary). SM3 does *not* require any external decay schedule. That said, all algorithms we experimented with benefited from a short warmup phase of a few thousand updates. This is currently detailed in the supplementary material, and in the final version it will be made explicit in the main text. Thank you for pointing out the lack of clarity. ● *AdaFactor for general tensors:* The AdaFactor paper did not describe an extension to higher order tensors, and indeed, the source code of AdaFactor breaks tensors into disjoint matrices (slices) each of which is approximated separately. Further, AdaFactor's approximations can be over or underestimates of the gradient moments, and in our experiments we see training instability which precluded the usage of AdaFactor with a batch size of 2048 (Figure 3 in the paper). SM3 does not suffer from this issue. Its *theoretical* and *empirical* convergence properties naturally extend to higher ranks. ● *Breakdown of memory usage:* We will definitely try to elaborate more on this important point in the final version, space permitting. We would appreciate it if you could add some details in your final review about what kind of breakdown you would have liked to see. ● *Analysis under different assumptions:* It would be indeed interesting to extend our analysis to various non-convex settings, and we are currently working in that direction. In terms of over-parameterization, is there a concrete assumption you think might improve convergence bounds? Please do add it to your final review, we would really appreciate it.

(a) Input embedding    (b) Attention layer    (c) Output softmax

Figure 1: Magnitude of the 100 largest accumulators of Adagrad and its approximation with SM3 and Count Sketch (SKMS'19) for a Transformer model trained on WMT'14 en→fr dataset.

[Meta-Review · NeurIPS 2019]

There was a divergence of opinions among the reviewers regarding this paper, and no consensus was reached. I am recommending acceptance of the paper, but recommend that the authors implement the suggested changes to the empirical evaluation in the final version.